# Region of Interest Detection in Melanocytic Skin Tumor Whole Slide Images—Nevus and Melanoma

**DOI:** 10.3390/cancers16152616

**Published:** 2024-07-23

**Authors:** Yi Cui, Yao Li, Jayson R. Miedema, Sharon N. Edmiston, Sherif W. Farag, James Stephen Marron, Nancy E. Thomas

**Affiliations:** 1Department of Economics, University of North Carolina at Chapel Hill, Chapel Hill, NC 27599, USA; yicui@unc.edu; 2Department of Statistics & Operations Research, University of North Carolina at Chapel Hill, Chapel Hill, NC 27599, USA; yaoli@email.unc.edu (Y.L.); marron@email.unc.edu (J.S.M.); 3Department of Pathology and Laboratory Medicine, University of North Carolina at Chapel Hill, Chapel Hill, NC 27599, USA; jayson_miedema@med.unc.edu; 4Lineberger Comprehensive Cancer Center, UNC School of Medicine, University of North Carolina at Chapel Hill, Chapel Hill, NC 27599, USA; edmiston@ad.unc.edu; 5Eshelman School of Pharmacy, University of North Carolina at Chapel Hill, Chapel Hill, NC 27599, USA; sherif_farag@med.unc.edu; 6Department of Biostatistics, University of North Carolina at Chapel Hill, Chapel Hill, NC 27599, USA; 7Department of Dermatology, UNC School of Medicine, University of North Carolina at Chapel Hill, Chapel Hill, NC 27599, USA

**Keywords:** deep learning, region of interest detection, melanocytic skin tumor, nevus, melanoma

## Abstract

**Simple Summary:**

Detecting key areas in tissue samples is crucial for accurate and timely cancer diagnosis. This study focuses on using advanced deep learning techniques to analyze images of skin tumors, specifically melanoma and nevi. By training our deep learning-based model on a dataset of these images, we aimed to detect key areas in each image and simultaneously classify whether a given image shows melanoma or a benign nevus. Our method achieved an accuracy of 92.3% in classifying the images from a test set, meaning it correctly identified most cases. This approach not only helps in distinguishing between these types of skin tumors but also shows promise for broader applications in diagnosing various cancers, potentially improving clinical outcomes and reducing healthcare costs.

**Abstract:**

Automated region of interest detection in histopathological image analysis is a challenging and important topic with tremendous potential impact on clinical practice. The deep learning methods used in computational pathology may help us to reduce costs and increase the speed and accuracy of cancer diagnosis. We started with the UNC Melanocytic Tumor Dataset cohort which contains 160 hematoxylin and eosin whole slide images of primary melanoma (86) and nevi (74). We randomly assigned 80% (134) as a training set and built an in-house deep learning method to allow for classification, at the slide level, of nevi and melanoma. The proposed method performed well on the other 20% (26) test dataset; the accuracy of the slide classification task was 92.3% and our model also performed well in terms of predicting the region of interest annotated by the pathologists, showing excellent performance of our model on melanocytic skin tumors. Even though we tested the experiments on a skin tumor dataset, our work could also be extended to other medical image detection problems to benefit the clinical evaluation and diagnosis of different tumors.

## 1. Introduction

The American Cancer Society predicted that in 2022 an estimated 99,780 cases of invasive and 97,920 cases of in situ melanoma would be newly diagnosed and 7650 deaths would occur in the US [1]. The state-of-the-art histopathologic diagnosis of a melanocytic tumor is based on a pathologist’s visual assessment of its hematoxylin and eosin (H&E)-stained tissue sections. However, multiple studies have suggested high levels of diagnostic discordance among pathologists in interpreting melanocytic tumors [2,3,4]. Correct diagnosis of primary melanoma is key for prompt surgical excision to prevent metastases and in identifying patients with primary melanoma who are eligible for systemic adjuvant therapies that can improve survival. Alternatively, overdiagnosis can lead to unnecessary procedures and treatment with toxic adjuvant therapies. We are applying deep learning methods in computational pathology to determine if we can increase diagnostic accuracy, along with increasing speed and decreasing cost. Here, we examine methods for improving region of interest (ROI) detection in melanocytic skin tumor whole slide images (WSIs), which is an important step toward the computational pathology of melanocytic tumors.

Traditionally, expert pathologists visually identify and annotate the potential or related regions for melanoma and nevus and then take a close look to classify certain types. However, this process is time-consuming and the accuracy is also not satisfactory [5]. One potential solution may be the combination of high-quality histopathological images and AI technology. Histopathological images have long been utilized in treatment decisions and prognostics for cancer. For example, histopathological images are used to score tumor grade in breast cancer to predict outcomes for cancer cases or perform histologic pattern classification in lung adenocarcinoma, which is critical for determining tumor grade and treatment for patients [6,7]. AI technology, like deep learning-based predictors trained on annotated and non-annotated data, could be a potentially efficient technology to improve early detection [8], help pathologists diagnose tumors and inform treatment decisions to potentially improve overall survival rates.

Recently, with the advancement of machine learning, especially deep learning, many researchers have developed various frameworks and Convolutional Neural Network (CNN) architectures, like ZefNet [9], Visual geometry group (VGG) [10], ResNet [11], DenseNet [12], etc., to solve the biomedical image computing and classification problems in the field of computer vision and pathology [6,7,13,14,15,16]. The idea of transfer learning from these frameworks is to use a network that has been trained on unrelated categories on a huge dataset, like Imagenet, and then transfer its knowledge to the small dataset. Besides these transfer learning-based methods, there also exist some methods that do not use the pre-trained model. These models are trained only by training datasets to update all the CNN parameters. In other words, deep learning largely expands our methods of dealing with prediction and classification problems in pathology, and the applications include tumor classification [17,18], cancer analysis and prediction [6,7], cancer treatment prediction [19,20] and so on. Therefore, in the field of computational pathology, more and more researchers use these deep learning methods on medical images that include rich information and features. Some papers [6,18,21] try to use WSIs and show high performance and accuracy of their models on certain types of cancers like breast cancers and uterine cancers. Recently, there has been some literature [22] analyzing skin cancer based on histopathological images. However, previous literature does not include the ROI detection of melanocytic skin tumors and only has limited accuracy in classification as well as identification among various tumors.

Benefiting from AI technology and histopathological images, we developed a deep neural network-based ROI detection method that could precisely detect the ROI in melanocytic skin tumors through WSIs and, at the same time, classify the slides accurately. In Figure 1, the slides have an ROI indicated by black dots. Our goal was to automatically find this region without the use of black dots. The performance of our model could be seen as the green boundary in the right panel. Large images were broken into small patches and we extracted abundant features from these patches [23,24,25,26,27]. In addition, we leveraged the partial information from annotations, also called “semi-supervised learning”, to enhance our model detection method. This method improved classification accuracy compared to previous approaches for certain kinds of tumors. Also, we proved our algorithm’s accuracy and robustness by decreasing our training samples to various subsets of the original training samples. Figure 2 illustrates an overview of our method.

## 2. Materials and Methods

### 2.1. Data

The Melanocytic Tumor Dataset contained 86 melanoma (skin cancer) and 74 nevi (benign moles) WSIs. (Each whole slide image can generate thousands of patches, resulting in a substantial increase in the effective training data available. Specifically, by extracting and utilizing patches from each H&E image, we aggregate a dataset comprising hundreds of thousands of patches. This approach not only enriches the training data but also enhances the model’s ability to generalize across different regions within each slide.) Besides slide-level labels, there were annotations made by pathologists on these slides. A slide might contain multiple slices of the same tissue, and pathologists annotated ROIs on some slices for diagnosis purposes, but not others. We used an Aperio ScanScope Console to scan the tissue samples with 20× magnification.

We randomly selected 80% (134 WSIs: it contained 71 melanoma (skin cancer) and 63 nevus (benign moles) WSIs) of the data as our training set (Figure 2a). For the training set, the slide-level labels (melanoma vs. nevus) are available, but the true annotations of ROI are not. While a portion of the ROIs in the slides were annotated, it should be noted that not all ROIs received annotation. This causes a challenge in using these annotations to evaluate the performance of the model in the ROI detection task. However, we can still leverage these partial annotations to train a deep learning model that can perform slide classification and ROI detection.

We took the other 20% (26 WSIs) as our testing set (Figure 2a). The primary aim of this split is to evaluate the generalizability of the model. This balance helps to mitigate the risk of overfitting. For the evaluation of our method and other baseline models, these 26 WSIs were manually annotated by our team. (This is a set of slides which was reviewed and the melanoma and nevi were circled on the glass slides by an expert dermatopathologist (PAG). All the slides are not deidentified.) Our model was trained on slides (from the training set) without ground-truth annotations with only partial information on WSIs. We used an Aperio ImageScope Console to mark tumor boundaries as annotations and exported these annotations from the aforementioned software in the Extensible Markup Language (XML) format. It also included the annotated regions related to corresponding coordinates. We utilized these coordinates for each slide to figure out these regions solely from the rest of the image, labeled as melanoma or nevus.

### 2.2. Data Preparation

*Data preprocessing: color normalization.* To minimize the potential side effects of color recognition, we preprocessed all WSIs using previous color normalization methods [28,29]. There were different qualities or colors for scans performed in different labs or even the same lab for scans of the same WSIs processed at different times. The model may detect these undesirable changes to influence feature extraction and even the following classification and ROI detection. Thus, we applied color normalization methods to these WSIs to ensure the slides that were processed under different circumstances were in the common, normalized space, which could enhance the robustness of model training and quantitative analysis (Figure 2b). *Data preprocessing: data augmentation.* First, tissue detection for the patches extracted from WSIs was completed. If we detected certain tissues, we would collect these tissues into patches and then finish the color normalization part. Data augmentation was then performed by randomcorp, random horizontalflip and normalization of patches (Figure 2b). And the edge features were restored accurately. *Patch extraction.* Image slides were tiled into non-overlapping patches of 256 × 256 pixels in 20× magnification. Given a WSI, patches were extracted based on the slide-level label and annotations (Figure 2c). If the slide-level label was nevus, all patches inside the annotated regions were labeled as nevus. If the slide-level label was melanoma, all patches inside the annotated regions were labeled as melanoma. Besides patches from annotated regions, some patches outside those regions were also extracted and labeled as other. However, since not all ROIs were annotated by pathologists, there could be melanoma and nevus patches outside annotated regions. To avoid labeling those patches as other, we manually extracted patches of other classes from regions.

### 2.3. Model Training and Assessment

A three-class patch classification model (PCLA-3C) was trained on the labeled patches with VGG16 [10] as the base architecture (Figure 2d). Models were trained using this CNN architecture, and by backpropagation, we manually changed the last layer’s parameters to optimize the model. The patch classifier would return a WSI with three key scores, corresponding to three categories (melanoma, nevus and other). In the testing stage, all patches from a WSI were first fed into the trained patch classifier. Ignoring patches predicted as other, slide-level prediction was performed by majority vote based on patches predicted as melanoma and nevus. If the number of patches labeled as melanoma exceeded the number of patches labeled as nevus in one WSI, we classified it as melanoma, and vice versa (Figure 2e). For a WSI classified as melanoma, all the patches from this slide will be ranked by melanoma predicted scores. Otherwise, all the patches will be ranked by nevus predicted scores (Figure 2f).

To evaluate the performance of ROI detection, the annotated ratio was measured to calculate Intersection over Union (IoU) for each slide. Given a slide, annotated ratio β was calculated by the number of patches in the annotated region divided by the number of patches extracted from the slide: $\beta =\frac{A_{p}}{C_{p}}$, where $A_{p}$ is the number of patches in *A* (annotated region) and $C_{p}$ is the number of patches in *C* (WSI). Then, the top nβ patches based on predicted scores were classified as ROIs, where *n* was the total number of patches from a slide. For example, if β=0.2 for a slide in the testing set, it means that 20% of the regions in the slide are ROIs. Then, the model will predict the top 20% of patches (based on the predicted scores) as patches in the ROIs. The performance was measured by Intersection over Union (IoU), which compared the annotated region and predicted ROI region. Since the framework was patch-based, IoU was calculated by the number of patches in the intersection region (the region in both annotated and predicted regions) divided by the number of patches in the union of the annotated and predicted ROI regions: $\mathrm{IoU}=\frac{{\underline{AB}}_{p}}{{\bar{AB}}_{p}}$, where ${\underline{AB}}_{p}$ shows the number of patches in the region of (A∩B) and ${\bar{AB}}_{p}$ shows the number of patches in the region of (A∪B). *A* is the annotated region and *B* is the predicted/highlighted region.

The detection methods could provide three types of visualization maps: boundary, overlap and heatmap. (The overlap map highlights the top-ranked patches in a WSI while masking other areas with a transparent blue color. The percentage of highlighted patches corresponds to β, the annotated ratio. Therefore, the highlighted region represents the predicted ROI. This visualization helps in identifying the most significant areas that the model considers as part of the ROI. The boundary map delineates the boundary of the largest ROI cluster based on the highlighted patches. The highlighted patches are clustered using the OPTICS algorithm, which effectively groups the patches into meaningful clusters. This map provides a clear visual representation of the boundaries of the detected ROI, aiding in the assessment of the model’s precision in identifying tumor margins. The heatmap uses a color gradient to indicate the predicted scores, with red regions representing high predicted scores and blue regions representing low predicted scores. This visualization allows for an intuitive understanding of the model’s confidence in different regions of the WSI, highlighting areas with high likelihoods of being part of the ROI.) Three visualization maps are generated based on the predicted scores calculated in the ROI detection section (Figure 2g). The overlap map highlights top-ranked patches in a WSI and masks other areas with a transparent blue color (Figure 3a,d). The percentage of highlighted patches equals β (the annotated ratio). Therefore, the highlighted region is also the predicted ROI. The boundary map shows the boundary of the largest ROI cluster based on the highlighted patches, where the highlighted patches are clustered by the OPTICS algorithm [30] (Figure 3b,e). The last one is a heatmap where red covers regions that have high predicted scores and blue covers regions that have low predicted scores (Figure 3c,f).

## 3. Results

### 3.1. Method Comparison

Two methods were tested on the Melanocytic Skin Tumor Dataset to conduct ROI detection and slide classification: (1) CLAM (clustering-constrained attention multiple instance learning) [31] and (2) PCLA-3C (the proposed patch-based classification model). The 160 WSIs from the UNC Melanocytic Tumor Dataset cohort were randomly split into training and testing sets with 134 for training and 26 for testing. Both methods were trained on the training set, and the performances on both the training and testing sets were evaluated. Visualization results and code can be found on GitHub (https://github.com/cyMichael/ROI_Detection, accessed on 16 July 2024).

All analyses were conducted using Python. Images were analyzed and processed using OpenSlide. All the computational tasks were finished on UNC Longleaf Cluster with Linux (tested on Ubuntu 18.04) and NVIDIA GPU (tested on Nvidia GeForce RTX 3090 on local workstations). NVIDIA GPU supports were followed to set up and configure CUDA (tested on CUDA 11.3), and the torch version should be greater than or equal to 1.7.1. Additionally, we acknowledge FastPathology [32] (https://github.com/AICAN-Research/FAST-Pathology, accessed on 17 July 2024) as an alternative method that could be applied alongside our Python-based framework. Both platforms are designed to be user-friendly, and incorporating FastPathology could provide additional flexibility and accessibility for users.

### 3.2. Model Validation and Robustness

We trained the model based on different proportions of the training dataset, but the results were based on the testing set (26 WSIs), see Table 1. There was a high agreement between the predictions of the ROI by PCLA-3C and the true ones, showing the accuracy of our automatic ROI detection.

In our study, we have designated nevi as the positive class and melanoma as the negative class. This choice aligned with clinical priorities, where correctly identifying benign cases (nevi) is critical to avoid unnecessary interventions. Table 2 shows the confusion matrix with accuracy (93.5%), sensitivity (81.8%) and specificity (100%). These metrics collectively demonstrate the effectiveness of our model in distinguishing between nevi and melanoma with a high degree of accuracy. The high specificity is particularly reassuring from a clinical perspective, ensuring that malignant cases are reliably identified.

By using the training data, our method achieved an accuracy of 92.3% in slide-level classification and IoU rate of 38.2% in the ROI detection task on the testing set. Our method achieved better accuracy than CLAM with an accuracy of 69.2% in slide-level classification and IoU rate of 11.2% in the ROI detection task. Also, we analyze the robustness results in the Supplementary Information, showing the accuracy was 0.7866 (95% CI, 0.761–0.813) at the patch level and the accuracy was 0.885 (95% CI, 0.857–0.914) at the slide level by using 80% (107 WSIs) of the original training set. Our true testing data were kept unchanged since these data included true annotations. However, the training data did not include the true annotations. As in the PCLA-3C, the improvements in patch classification accuracy, slide classification accuracy and IoU showed the importance of annotations in the training of deep learning classifiers for prediction. Also, we showed that patch classification results can be used to predict the slide-level label accurately. This is important as accurate tumor type is a clinical biomarker for future treatment. In summary, our deep learning-based framework has outperformed the state-of-the-art ROI detection method [31], leading to better model visualization and interpretation. This is quite crucial in medical imaging fields and related treatment recommendations.

### 3.3. Misclassified Slides Discussion

The proposed method PCLA-3C only misclassified two slides in the testing set. The two WSIs were both labeled as nevus but misclassified as melanoma by the model (see the two slides and corresponding visualization results in Figure 4 and Figure 5). The slide in Figure 4 is not a typical nevus and it has the features of a pigmented spindle cell nevus, which is one diagnostic challenge of melanocytic skin tumors. However, the slide in Figure 5 is a routine type of nevus. The reason that PCLA-3C misclassified the slide could be based on the difference in color. In general, the ROIs in melanoma cases were dark, while those in nevus cases were light. As shown in Figure 5b, there were some dark areas outside the annotated ROIs, which contributed to the misclassification of slides and the incorrect detection of ROIs.

## 4. Discussion

In this work, we presented deep learning-based classifiers for predicting correct tumor types with and without annotations. Using high-quality WSIs from the UNC Melanocytic Tumor Dataset cohort annotated by our pathologists, we systematically selected the proper cases for training and testing. Heatmap, boundary and overlay figures exerted by PCLA-3C showed a considerable agreement with annotations finished by our pathologist group. Also, as shown in Table 1 and Table 3, the test results showed that PCLA-3C had higher accuracy at the patch level, slide level and ROI level by just using limited WSIs as the training set than CLAM.

Some recent studies have also examined tumors by using deep learning architecture in the medical imaging field [13,14,15,16,33]. Most literature mainly studied the effects of CNN-based methods on different cancers like breast cancer and skin cancer and achieved high accuracy on the classification task. The main difference between our model and those CNN-based models is that our model surpasses the state-of-the-art (SOTA) performance of the CLAM model. Our approach achieves higher accuracy in dealing with medical images, particularly in the context of melanocytic skin tumors, demonstrating superior performance in both detection and classification tasks. In addition to CNN-based methods, we have also included recent research focusing on Transformer models, which have demonstrated powerful learning capabilities in various medical image enhancement tasks. For instance, Feng et al. [34] introduced an end-to-end task Transformer network (T2Net) that allows feature representations to be shared and transferred between MRI reconstruction and super-resolution tasks. Similarly, Wang et al. [35] introduced the Transformer-based TED-net for low-dose CT (LDCT) denoising. TED-net is an encoder–decoder dilation network free of convolution, utilizing a symmetric U-shaped architecture with encoder–decoder blocks consisting solely of the transformer. Khalid et al. [36] utilized deep learning and transfer learning to classify skin cancers. Some literature [37,38] tried to solve the classification problem in breast cancer by deep learning methods. In addition, Farahmand et al. [6] not only classified the WSI accurately, but they also focused on ROI detection tasks and achieved nice results. From Lu et al. [31], CLAM was used to solve the detection of renal cell carcinoma and lung cancer. CLAM is proposed to accomplish slide classification and ROI detection, which does not require pixel- or patch-level labels. However, when applied to the Melanocytic Skin Tumor Dataset, the ROI detection of this method is not satisfactory. Lerousseau et al. [39] introduced a weakly supervised framework (WMIL) for WSI segmentation that relies on slide-level labels. Pseudolabels for patches were generated during training based on predicted scores. Their proposed framework has been evaluated on multi-locations and multi-centric public data, which demonstrates a potentially promising approach for us to further study WSIs.

Here, we reported on a novel method that performed automated ROI detection on primary skin cancer WSIs. It improved the performance of the state-of-the-art method by a large margin.

In most places, diagnostic pathologists will manually scan all the slides to analyze tumor types. Thus, it is convenient and cheap to apply a deep learning method to these existing WSIs. The high accuracy of our deep learning-based method results has made huge progress toward digital assistance in diagnosis.

The key strength of our model is that it overcomes the lack of ground-truth labels for the detection task. The performance of previous methods was not satisfactory on melanocytic WSIs. One reason is that melanocytic tumors are difficult to diagnose and detect, and the literature reports 25–26% of discordance between individual pathologists for classifying a benign nevus versus malignant melanoma [40]. Using only slide-level labels was hard in training a promising method. The success of our method means that the combination of partial information from annotations and patch-level information could largely enhance the analysis of melanocytic skin tumors.

The weakness of our model is that our model does not classify all the WSIs accurately. Our slide classification is 92.3%, so we could not rely completely on the model (PCLA-3C). Two WSIs (true label: nevus) in the testing set were misclassified as melanoma. Although our method does not perform the same as the gold standard, our results can assist pathologists in efficiently classifying WSIs and finding the ROI.

## 5. Conclusions

In summary, the deep learning architecture that we developed and utilized in this study could produce a highly accurate and robust approach to detect skin tumors and predict the exact type of tumors. Given that it takes lots of time to examine patients’ WSIs, besides the conventional methods, our efficient AI method could help medical staff save time and improve the efficiency and accuracy of diagnosis, which benefits each patient in the future. We expect that our approach will be generalizable to other cancer-related types, not restricted to skin cancer, or breast cancer [7] and vision-related treatment outcome predictions. The deep learning-based framework could also be widely applied in identification and prediction in diagnostics. In the future, we plan to extract some detailed information from high-quality WSIs and then improve our model to obtain higher accuracy in detection and prediction. Future work will also include further improvements in ROI detection performance by incorporating extra information into the model, such as gene expression and clinical data.

## Figures and Tables

**Figure 1 cancers-16-02616-f001:**
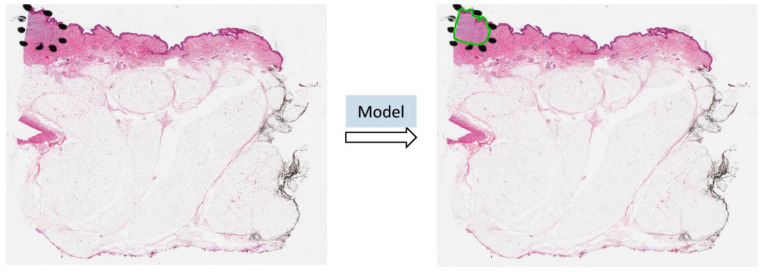
The ROI was annotated by black dots determined by pathologists. The predicted ROI was bounded by the green line on the right.

**Figure 2 cancers-16-02616-f002:**
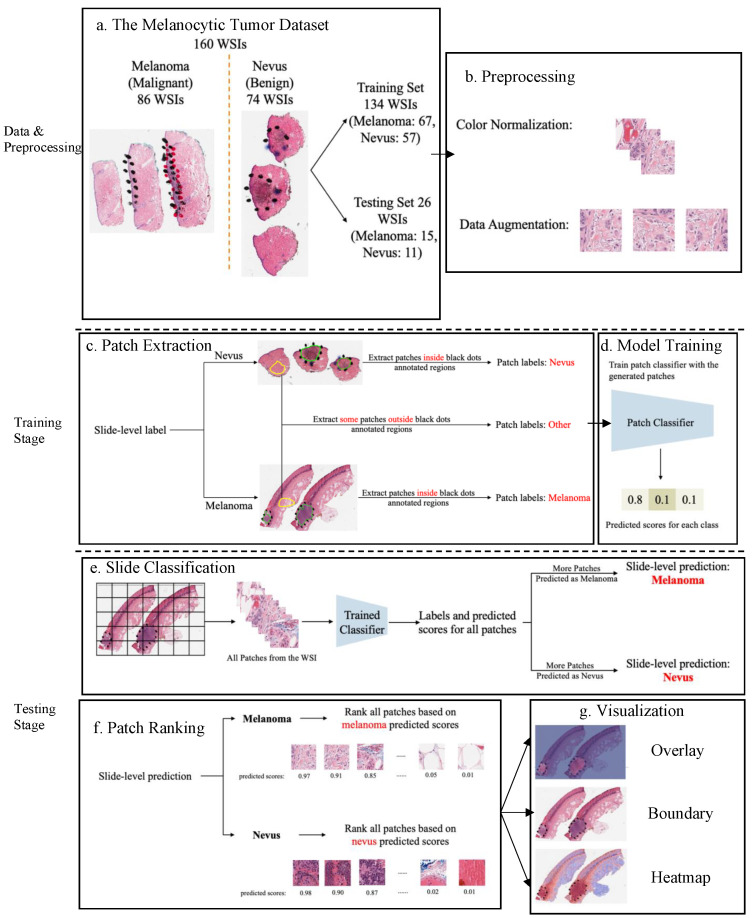
An overview of the proposed detection framework. (**a**) The Melanocytic Tumor Dataset randomly assigned 80% (134 WSIs) of the data as the training set and 20% (26 WSIs) of the data as the testing set. (**b**) Preprocessing: color normalization [28,29] and data augmentation. (**c**) Extract melanoma, nevus and other patches from training data. (**d**) Model trained a 3-class patch classifier based on extracted patches. (**e**) For each slide, slide classification generated predicted scores for all patches and calculated patch and slide classification accuracy. (**f**) All patches from a slide were ranked based on the corresponding predicted scores in the context of melanoma or nevus, depending on the slide classification result. (**g**) Visualization results based on predicted scores.

**Figure 3 cancers-16-02616-f003:**
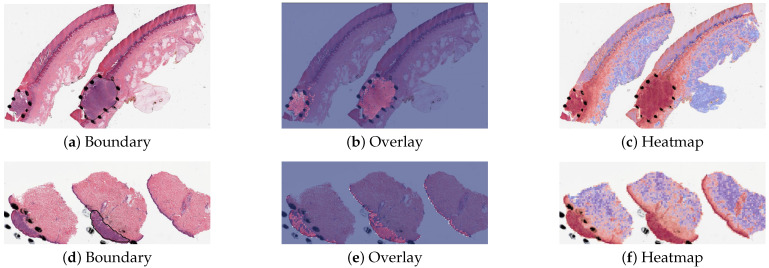
Visualization results for a melanoma sample and a nevus sample.

**Figure 4 cancers-16-02616-f004:**
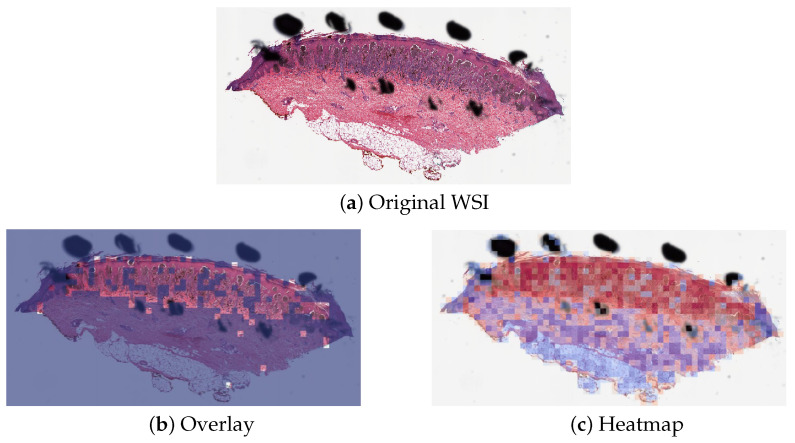
Visualization results for misclassified case 1.

**Figure 5 cancers-16-02616-f005:**
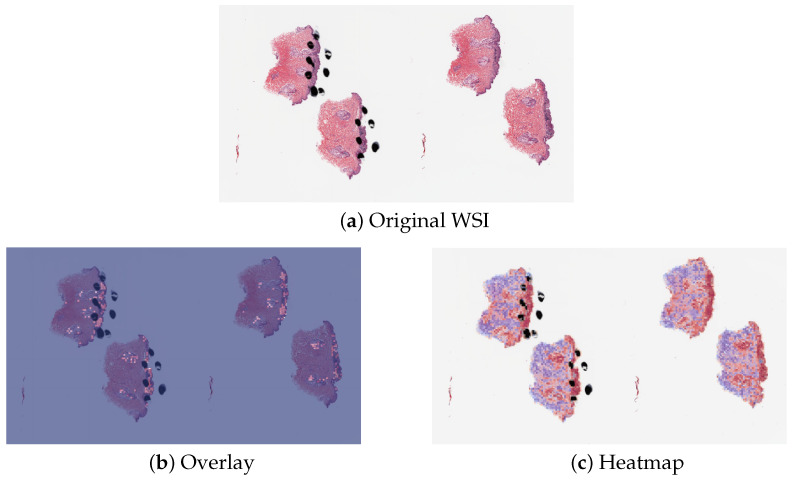
Visualization results for misclassified case 2.

**Table 1 cancers-16-02616-t001:** Performance of patch classification accuracy, slide classification and IoU by PCLA-3C and CLAM using the original training set.

Evaluation Metrics	PCLA-3C	CLAM
Patch classification accuracy	0.892	-
classification accuracy	0.923	0.692
IoU	0.382	0.112

**Table 2 cancers-16-02616-t002:** Confusion matrix by PCLA-3C using the original training set.

	True: Nevi	True: Melanoma
Predicted: Nevi	20	0
Predicted: Melanoma	2	9

**Table 3 cancers-16-02616-t003:** The robustness performance of patch classification accuracy, slide classification and IoU by PCLA-3C and CLAM using different splits of the original training set. Since CLAM does not perform patch classification, it does not have patch classification accuracy.

	20% Split				40% Split			
	PCLA-3C		CLAM		PCLA-3C		CLAM	
	Mean	95% CI	Mean	95% CI	Mean	95% CI	Mean	95% CI
Patch classification accuracy	0.6397	[0.5193, 0.7601]	-	-	0.7887	[0.7536, 0.8238]	-	-
Slide classification accuracy	0.7406	[0.6627, 0.8185]	0.6710	[0.6386, 0.7033]	0.8430	[0.8043, 0.8817]	0.6976	[0.6619, 0.7333]
Intersection over Union	0.3026	[0.2394, 0.3327]	0.0427	[0.0342, 0.0512]	0.3402	[0.3057, 0.3784]	0.0524	[0.0297, 0.0751]
	60% split				80% split			
	PCLA-3C		CLAM		PCLA-3C		CLAM	
	Mean	95% CI	Mean	95% CI	Mean	95% CI	Mean	95% CI
Patch classification accuracy	0.8191	[0.7766, 0.8616]	-	-	0.8210	[0.7949, 0.8471]	-	-
Slide classification accuracy	0.8721	[0.8458, 0.8985]	0.7097	[0.6830, 0.7364]	0.8885	[0.8607, 0.9163]	0.7258	[0.7117, 0.7399]
Intersection over Union	0.3652	[0.3369, 0.3934]	0.0621	[0.0428, 0.0814]	0.3710	[0.3335, 0.4084]	0.1103	[0.0529, 0.1677]

## Data Availability

The datasets used and/or analyzed during the current study are available from the corresponding author on reasonable request. All data generated or analyzed during this study are included in this paper [41].

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
