# Peer review of "Region of Interest Detection in Melanocytic Skin Tumor Whole Slide Images—Nevus and Melanoma"

_cancers, 2024, doi:10.3390/cancers16152616_

Round 1

Reviewer 1 Report

Comments and Suggestions for Authors

The manuscript is of interest and overall well-written. There are few minor issues to be clarified:

The dataset was randomly divided into a section for training involving 80% of images and the remaining 20% was assigned to testing: why such a discrepancy was present? Please explain the rationale and add a power analysis for further validation.

- The study would benefit from adding the estimates of sensitivity, specificity and AUC.

Author Response

Dear Reviewer,

Thank you for your valuable feedback. Additionally, please find attached the updated manuscript with our detailed response in the PDF file.

Thank you once again for your constructive comments.

Sincerely,
Yi

Reviewer 2 Report

Comments and Suggestions for Authors

The authors need to address the following concerns:

1. The dataset contains a very small number of images, with only 160 hematoxylin and eosin images, which seems insufficient for training a deep learning model.

   2. The specific deep learning model utilized by the authors is not clearly mentioned.

3. There are no details provided about the experts who performed the manual annotation of the ROI by pathologists.

4. The authors need to provide the structure of the utilized deep learning model.

5. Training curves of the deep learning models have not been provided.

6. Performance metrics validations of the proposed methodology need to be provided.

7. The literature review is inadequate, requiring more research articles and a clear identification of the research gap.

8. A state-of-the-art table comparing the outcomes of the proposed techniques with existing methods should be provided.

9. There are no significant contributions evident from the authors' work; substantial improvements are needed. 

Comments on the Quality of English Language

The manuscript is poorly written, with many flaws in sentence formation, grammatical errors, typos, spelling mistakes, and phrasing issues. These need to be corrected.

Author Response

(The authors gave the same response as above.)

Reviewer 3 Report

Comments and Suggestions for Authors

The authors presented the work titled as "Region of Interest Detection in Melanocytic Skin Tumor Whole Slide Images - Nevus & Melanoma" where they used deep learning approach to visualize and analyze the ROI (region of interest). The overall work sounds novel and well planned. I have few technical concerns:

In detection framwork:

1. The framework is in four stages: data processing, training, slide classification, and test stage. Color normalization and data augmentation are the two sub-stages of data processing. I feel that it would be more worth if the authors will discuss it in more details and describe about the exact segmentation also. How is this framework different from FastPathology?

2. At testing stage overlay and boundary needs more clarity in terms of method and explanation.

Author Response

(The authors gave the same response as above.)

Round 2

Reviewer 2 Report

Comments and Suggestions for Authors

Authors have made the required modifications, may be accepted for publication.